# Coordinated Regulation of Central Carbon Metabolism in Pyroligneous Acid-Treated Tomato Plants under Aluminum Stress

**DOI:** 10.3390/metabo13060770

**Published:** 2023-06-20

**Authors:** Raphael Ofoe, Raymond H. Thomas, Lord Abbey

**Affiliations:** 1Department of Plant, Food, and Environmental Sciences, Faculty of Agriculture, Dalhousie University, 50 Pictou Road, Bible Hill, NS B2N 5E3, Canada; 2Department of Biology, Faculty of Science, Western University 2025E Biological & Geological Sciences Building, 1151 Richmond Street, London, ON N6A 5B7, Canada; rthoma2@uwo.ca

**Keywords:** wood vinegar, metabolomics, TCA cycle, carbohydrate metabolism, *Solanum lycopersicum*

## Abstract

Aluminum (Al) toxicity is a major threat to global crop production in acidic soils, which can be mitigated by natural substances such as pyroligneous acid (PA). However, the effect of PA in regulating plant central carbon metabolism (CCM) under Al stress is unknown. In this study, we investigated the effects of varying PA concentrations (0, 0.25 and 1% PA/ddH_2_O (*v*/*v*)) on intermediate metabolites involved in CCM in tomato (*Solanum lycopersicum* L., ‘Scotia’) seedlings under varying Al concentrations (0, 1 and 4 mM AlCl_3_). A total of 48 differentially expressed metabolites of CCM were identified in the leaves of both control and PA-treated plants under Al stress. Calvin–Benson cycle (CBC) and pentose phosphate pathway (PPP) metabolites were considerably reduced under 4 mM Al stress, irrespective of the PA treatment. Conversely, the PA treatment markedly increased glycolysis and tricarboxylic acid cycle (TCA) metabolites compared to the control. Although glycolysis metabolites in the 0.25% PA-treated plants under Al stress were comparable to the control, the 1% PA-treated plants exhibited the highest accumulation of glycolysis metabolites. Furthermore, all PA treatments increased TCA metabolites under Al stress. Electron transport chain (ETC) metabolites were higher in PA-treated plants alone and under 1 mM, Al but were reduced under a higher Al treatment of 4 mM. Pearson correlation analysis revealed that CBC metabolites had a significantly strong positive (r = 0.99; *p* < 0.001) association with PPP metabolites. Additionally, glycolysis metabolites showed a significantly moderate positive association (r = 0.76; *p* < 0.05) with TCA metabolites, while ETC metabolites exhibited no association with any of the determined pathways. The coordinated association between CCM pathway metabolites suggests that PA can stimulate changes in plant metabolism to modulate energy production and biosynthesis of organic acids under Al stress conditions.

## 1. Introduction

Soil acidity is widespread and accounts for 50% of the global agricultural lands that support up to 80% of world vegetable production [1]. Aluminum (Al) is the most abundant metal in the lithosphere, and its availability is dependent on soil acidity levels [2]. Al exists in soils as non-toxic oxides and Al-silicates, to which several plant roots are exposed. However, when soil pH drops below 5, Al dissociates into trivalent forms that are toxic to most plants, including tomato, even at low concentrations [2]. These sensitive plants exhibit a wide range of Al-induced phytotoxicity, with the inhibition of root growth being the most significant effect [2,3]. Al disrupts root cells, leading to physiological drought by restricting water and nutrient uptake [4]. It also reduces leaf elongation, impairs the photosynthetic ability of plants and instigates the accumulation of reactive oxygen species (ROS) [5,6]. Al-induced buildup of ROS facilitates membrane lipid peroxidation, loss of cell membrane integrity and damage to cellular components including nucleic acids and proteins [7]. Consequently, these phytotoxic effects result in marked yield reduction and total crop failure [4,8].

Generally, plants have evolved cellular and metabolic mechanisms to mitigate Al-induced phytotoxicity. The exclusion and internal detoxification mechanisms of Al tolerance have been well-characterized in plants [2,4]. The exclusion mechanism involves preventing Al entry into root cells by secreting organic acids (OA) to chelate Al within the rhizosphere. The internal tolerance mechanism involves intracellular detoxification of Al and sequestration of the non-toxic Al–OA complexes in the vacuole [4]. These key mechanisms of Al tolerance can be linked to the central carbon metabolism (CCM) and its associated mitochondrial activity, but they are understudied. Furthermore, plant exposure to stress requires a rapid metabolic shift to maintain appropriate metabolism. As a result, several regulatory mechanisms must interact with various metabolic routes to regulate fluxes through the different intermediate pathways associated with CCM [9]. The CCM pathway supplies the fundamental carbon needed to sustain growth and productivity under stress conditions and comprises the Calvin–Benson cycle (CBC), glycolysis, pentose phosphate pathway and tricarboxylic acid (TCA) cycle, followed by oxidative phosphorylation for ATP production through the electron transport chain (ETC) [9,10,11]. Recently, several studies have demonstrated that Al stress alters metabolic profiles in plant, which may enhance Al tolerance [12,13,14]. Although most of these studies have focused on the root metabolism of Al tolerance, the effects of Al stress on metabolic changes in aerial parts or plants are poorly understood.

Mitochondrial metabolism plays an important role in Al tolerance by facilitating carbohydrate consumption for OA and cellular ATP biosynthesis [2,8]. Consequently, the glycolytic influx is increased under stress conditions to support not only energy production but also OA generation during Al detoxification. Notably, the TCA cycle produces high malic and citric acids in response to Al stress. These organic acids bind to Al ions, reducing Al toxicity both in the intracellular environment and around the rhizosphere [2]. Therefore, the end products of photosynthesis are highly utilized for OA generation during Al stress [6,12,15], which suggests that Al tolerance is linked to improved photosynthesis. Similarly, leaf metabolism under Al stress represents an important characteristic of plant growth and productivity, and the identification of metabolic alterations modulating photosynthesis and respiration may contribute to improving Al tolerance in plants.

Recent research and development have focused on the use of novel strategies to improve plant growth and resilience due to the multicomplex interaction of various environmental stresses [16]. As a result, the use of biostimulants to enhance crop growth, productivity and tolerance to environmental stresses has attracted the interest of both researchers and farmers [17,18]. Pyroligneous acid (PA) is one widely used biostimulant that is known to improve crop growth, resilience to environmental stresses and productivity [19,20]. PA is a translucent reddish-brown liquid produced through the carbonation of organic biomass in the presence of limited oxygen [19]. It consists of a complex mixture of over 200 water-soluble bioactive compounds, including organic acids, phenolics, sugar and alcohol derivatives [19,21,22]. However, its chemical composition depends on the feedstock, temperature, residence time and heating rate. In agriculture, PA has been demonstrated to promote seed germination, vegetative growth and yield in several crops [3,19,23].

Accumulating evidence has revealed that seed priming with PA treatments enhances plant tolerance to Al and drought stress by regulating genes and proteins involved in energy production and antioxidant defense systems [3,19,23]. Under drought stress, Wang et al. [23] reported that PA stimulates the differential accumulation of proteins involved in carbohydrate metabolism in wheat (*Triticum aestivum* L.) seedlings. This increase in drought tolerance was related to increased activities of enzymes associated to glycolysis and the TCA cycle. Similarly, Ofoe et al. [3] showed that PA enhances peroxidase activities and promotes the expression of auxin response factor and antioxidant genes in primed seedlings under Al stress, thus promoting Al tolerance. These findings suggest that PA can promote normal cellular balance and metabolic processes under stress conditions but require further investigation. Despite the accumulating efforts to study the simulation of plant growth and stress tolerance by PA, the metabolic mechanism underlying plant response to PA and Al stress remains unknown. Furthermore, no studies to date have explored the impact of PA on CCM regulation in plants under Al stress. In this study, we examined the metabolic profile involved in the CCM of PA-treated Scotia tomato seedlings under varying Al stress conditions. This investigation provides metabolic insights governing biochemical pathways that are crucial for understanding how PA mediates Al tolerance in plants.

## 2. Materials and Methods

### 2.1. Plant Material and Experimental Conditions

The experiment was conducted from January to March 2022 and repeated from March to May 2022 at the Department of Plant, Food, and Environmental Sciences, Faculty of Agriculture, Dalhousie University, Truro. Scotia tomato seeds were purchased from Halifax seeds (Halifax, Canada), and PA derived from white pine (*Pinus strobus*) biomass was produced by Proton Power Inc (Lenoir City, USA). Details of the PA and its chemical constituents were previously published by Ofoe et al. [21].

The Scotia tomato seeds were initially sterilized with 10% NaClO for 10 min and thoroughly washed 3 times with sterile distilled water. The seeds were further sterilized with 70% ethanol for 5 min and washed thoroughly with sterile distilled water. The sterilized seeds were sown in a Pro-Mix^®^ BX (Premier Tech Horticulture, Rivière-du-Loup, QC, Canada) and grown for four weeks in a growth chamber (Conviron Controlled Environments Ltd., Winnipeg, MB, Canada) with 16/8 h day/night photoperiod, 24/22 °C day/night temperature regime, 300 μmol m^−2^ s^−1^ light intensity and a relative humidity of 70%. Uniform seedlings with an 8-cm root length at the 3rd to 4th true leaf stage were transplanted into a 10.2-cm plastic pot containing 500 g of sterilized sand with an average particle size of 0.5–1.0 mm. The seedlings were maintained with a 25% strength Hoagland nutrient solution (pH = 5.0) at planting for a week under a growth chamber to acclimatize.

### 2.2. Experimental Treatment and Design

After one week of acclimation, half-strength followed by full-strength nutrient solutions were amended with varying PA and Al concentrations and applied every week. PA treatments were applied to the nutrient solution at 0%, 0.25% and 1% PA/ddH_2_O (*v*/*v*), with Al (AlCl_3_) concentrations of 0, 1 and 4 mM. The nutrient solution (pH = 4.5) was renewed every 3 days to maintain adequate moisture content. Throughout the entire study period, the pH of the amended nutrient solution (PA with or without Al) was monitored frequently and adjusted to 4.5 using either sodium hydroxide (NaOH) to increase the pH or HCl to reduce the pH. The study was arranged in a 3 × 3 factorial completely randomized design with five replications.

### 2.3. Plant Sample Preparation

The fully expanded leaf tissues on the third and fourth petioles from the top (15 leaves per treatment) were collected 40 days after transplanting and immediately frozen in liquid nitrogen. The frozen samples were ground into a fine powder and stored in a −80 °C freezer for central carbon metabolite analyses.

### 2.4. Metabolite Quantitation Using LC-MRM/MS

Targeted metabolite quantitation was performed at the University of Victoria Genome BC—Proteomics Centre of The Metabolomics Innovation Centre, Canada. Ground samples (50 mg) were mixed with 80% methanol and homogenized with two metal balls on an MM400 mill mixer (Retsch, Haan, Germany) for 3 min at a shaking frequency of 30 Hz. The mixture was then sonicated in an ice-water bath for 5 min and centrifuged at 12,000× *g* at 5 °C for 20 min. Next, 250 µL of the supernatant was mixed with 150 μL of water and 150 μL of dichloromethane. The mixture was vortexed for 30 s and centrifuged at 21,000× *g* for 20 min. Subsequently, 80 µL aliquots of the supernatant were dried under a nitrogen gas flow, and the residues were used for the following assay.

#### 2.4.1. TCA Cycle

A standard solution of all targeted carboxylic acids was prepared using 80% methanol, with concentrations ranging from 200 to 1000 μM. For each sample, 50 µL of both the standard solution and the sample supernatant were mixed with an equal volume of 200 mM 3-nitrophenyl hydrazines (NPH) solution and 150 mM carbodiimide hydrochloride (EDC)-6% pyridine solution. The mixture was then incubated at 30 °C for 40 min. After the reaction, 450 µL of water was added to each solution, and 10 µL of the resulting solution was injected into a C18 liquid chromatography (LC) column (2.1 × 100 mm, 1.8 μm) for quantification of carboxylic acids using ultrahigh LC-multiple reaction monitoring/mass spectroscopy (UPLC-MRM/MS) with (−) ion detection. The UPLC-MRM/MS was performed on an Agilent 1290 UHPLC system coupled to a Sciex 4000 QTRAP MS instrument (AB Sciex, Concord, ON, Canada). The metabolite-dependent parameters used in the UPLC-MRM/MS were based on the procedure described by Han et al. [24].

#### 2.4.2. Glucose and Selected Sugar Phosphates

The dried residue of each sample was mixed with 50 µL of 50% methanol. Then, 50 µL of each serially diluted standard solution of glucose, ribose, ribose-5-phosphate, glucose-6-phosphate and mannose-6-phosphate were mixed with 100 µL of 25 mM 3-amino-9-ethyl carbazole (AEC), 50 µL of 50 mM sodium cyanoborohydride (NaCBH_3_) and 20 µL of LC/MS grade acetic acid. The mixtures were incubated at 60 °C for 70 min, and 200 µL of water and 300 µL of chloroform were added. Following centrifugation at 12,500× *g* for 5 min, 50 µL of each supernatant was mixed with an equal volume of water, and 10 μL of the resulting solution was injected into a pentafluorophenylpropyl (PFP) LC column (2.1 × 150 mm, 1.7 μm) for UPLC-MRM/MS analysis. The UPLC-MRM/MS analysis was performed on an Agilent 1290 UHPLC system coupled to an Agilent 6495B QQQ instrument (Conquer Scientific, Poway, CA, USA) with positive-ion detection. The metabolite-dependent parameters used in the UPLC-MRM/MS were based on the procedure described by Han et al. [25].

#### 2.4.3. Other Metabolites

An internal standard (IS) solution containing 25 isotope-labelled metabolites, including adenosine diphosphate (ADP), adenosine triphosphate (ATP), fructose-6-phosphate (fructose-6P), fructose-bisphosphate, glycerol-3-phosphate, nicotinamide adenine dinucleotide (NAD), NADH, glucose-1-phosphate, ribose-5-phosphate and others, was prepared in 50% methanol with concentrations ranging from 0.00002 to 10 μM. The dried residue of each sample was dissolved in 100 µL of the IS solution. Then, 10 µL of the resulting solution and standard solutions were injected into a C18 LC column (2.1 × 100 mm, 1.9 μm) for UPLC-MRM/MS analysis with (−) ion detection. The UPLC-MRM/MS analysis was performed on a Waters Acquity UPLC system coupled to a Sciex QTRAP 6500 Plus MS instrument. A tributylamine acetate buffer—acetonitrile/methanol (1:1, *v*/*v*) was used as the mobile phase for gradient elution (10% to 50% B over 25 min) at a flow rate of 0.25 mL/min and a temperature of 60 °C.

### 2.5. Data Analysis

The concentrations of the detected analytes in the above assays were calculated using internal standard calibration. This involved interpolating the constructed linear regression curves of individual compounds using the analyte-to-internal standard peak area ratios measured from injections of the sample solution. The data analysis of the metabolites was performed using Analyst 1.6.2. A multivariate statistical analysis including two-dimensional (2D) principal component analysis (PCA) and hierarchical clustering was performed to assess the differential metabolism per group. The analysis was performed using XLSTAT version 2022.3 (Addinsoft, New York, NY, USA), and Euclidean distance was utilized for constructing the hierarchical clustering analysis.

## 3. Results and Discussion

### 3.1. Overall Metabolic Changes in PA-Treated Plants under Al Stress

Metabolites are the end products of cellular processes, and their levels are influenced by coordinated responses of biological systems to changes in internal and external conditions [26]. Although the metabolic analysis of plant responses to both PA and Al stress is relatively unknown, this study provides a comprehensive understanding of the complex metabolic changes that occur in PA-treated tomato plants under Al stress. Our cluster analysis of both PA-treated and control plants under Al stress revealed two main groupings based on the global metabolic profile (Figure 1B). While the effect of PA was somewhat comparable to the control group, distinct metabolic changes were observed with 1% PA under 1 mM Al stress, but not with 0.25% PA and 1 mM Al alone (Figure 1A,B). Moreover, the clustering of the control group, PA alone, 0.25% PA under 1 mM Al and 1 mM Al alone treatments indicated that the effect of 1 mM Al was less toxic to the tomato plants and resulted in a stable metabolic profile. In contrast, a considerable alteration in metabolic profile was observed in PA-treated and control groups under 4 mM Al stress, with the 1% PA treatment exhibiting the most pronounced effect (Figure 1A,B). These results were expected, as previous studies have shown that high PA and Al concentrations induce phytotoxic effects, while lower concentrations stimulate plant growth and productivity [12,20,27,28,29].

### 3.2. Differential Accumulation of Metabolites in PA-Treated Plants under Al Stress

Exposure of plants to adverse environmental conditions, including Al stress, necessitates rapid metabolic reprogramming to stimulate stress tolerance. These metabolic changes are coordinated among diverse pathways to alter fluxes related to the different CCM routes [9]. Table 1 shows the total differential abundance of metabolites involved in specific pathways within the CCM. In total, 48 metabolites were detected (Appendix A) and categorized into five groups based on their biological function in the CCM routes: CBC, glycolysis, PPP, TCA cycle and ETC (Table 1). Among these groups, glycolysis was the main differentially expressed metabolic pathway, accounting for approximately 75% of the total identified metabolites (Appendix A). Additionally, the TCA cycle accounted for approximately 25%, while the remaining groups comprised less than 1% of the total differentially expressed metabolites (Appendix A).

#### 3.2.1. Calvin–Benson Cycle

The Calvin–Benson cycle is the main biochemical pathway for carbon fixation in the chloroplast stroma of C3 plants [30]. During this process, plants utilize light energy to synthesize sugars and other carbon intermediates [31]. The regulation of CBC under stressful environments has been identified as a survival strategy for plants [31]. In the present study, both PA application and Al stress greatly affected the total CBC metabolites in the leaves of tomato seedlings (Table 1). PA treatments slightly reduced the total CBC metabolites compared to the control (Table 1). Similarly, Al stress exhibited a phytotoxic effect on CBC metabolites, especially under 4 mM Al, and this effect was further pronounced in PA-treated plants, suggesting that PA treatment could lead to decreased photosynthetic capacity under Al stress (Table 1). Moreover, the reduction in CBC metabolites under 4 mM Al stress can be attributed to low levels of all CBC metabolites except for sedoheptulose-7P (Figure 2A). This result agrees with previous studies that demonstrated a decrease in carbon fixation in tomato [32], spinach (*Spinacia oleracea*) [33], citrus [5,34,35] and rice (*Oryza sativa*) seedlings [36] under Al stress. Similarly, the decrease in total CBC metabolites caused by 0.25% PA, irrespective of Al exposure, may be attributed to low contents of all CBC metabolites except for DHAP, ribulose-5P and sedoheptulose-7P. While 3-phosphoglyceric acid, DAP, fructose 1,6-bisP and ribulose-bisP were increased in the 1% PA condition under 4 mM Al, the reduction in total CBC metabolites in 1% PA-treated plants under Al stress could be due to decreased glyceraldehyde-3P, fructose 6-phosphate, sedoheptulose-bisP, sedoheptulose-7P, ribose-5P and ribulose-5P (Figure 2A). Although PA treatment led to a reduction in total CBC metabolites, the impact of PA under Al stress could be considered slightly lower compared to 4 mM Al alone.

Previous research has suggested that the Al-induced downregulation of CBC results from both stomatal and non-stomatal factors [31,34]. Stomatal closure is thought to reduce leaf CO_2_ influx, which further impairs CBC processes [31]. However, some studies have indicated that stomatal closure alone cannot fully explain the Al-induced reduction in CBC. This is because intracellular CO_2_ levels were higher in Al-stressed leaves or comparable to those of control leaves, irrespective of stomatal conductance [5,6,32,34,37]. Pereira et al. [38] also indicated that Al-induced reduction in CBC was associated with thylakoid structural damage in lemon (*Citrus limon*) seedlings. Although the activities of CBC enzymes were not examined in this study, several studies have reported that heavy metal stress inhibits the activities of CBC enzymes, leading to a reduction in the overall CBC process [39,40,41,42]. Hence, the decrease in total CBC metabolites in both control and PA-treated plants under Al stress could be due to Al-induced structural damage and reduction in CBC enzyme activities [38,40].

#### 3.2.2. Glycolysis

Balanced energy flow under environmental stress plays a crucial role in plant growth, development and stress tolerance. Glycolysis is a significant catabolic pathway that breaks down carbohydrates to provide energy for physiological and cellular operations in plants [9]. In this study, carbohydrate catabolism via the glycolysis pathway was higher in 1% PA-treated plants, irrespective of Al treatment (Table 1). In the absence of Al exposure, the total glycolysis metabolites were markedly increased in the 1% PA-treated plants by approximately 0.75-fold compared to the control. When exposed to Al stress, glycolysis metabolites were sightly downregulated in the control plants, while the 1% PA treatment increased the total glycolysis metabolites by approximately 2-fold and 1.75-fold compared to the 1 mM Al and 4 mM Al treatments, respectively (Table 1). Moreover, the increase in total glycolysis metabolites with 1% PA treatment in the absence of Al exposure was due to high glucose and pyruvate acid contents (Figure 2B). On the other hand, the increase in total glycolysis metabolites in 1% PA-treated plants under Al stress can be ascribed to high levels of glucose, fructose 1,6-phosphate, DHAP, 3- and 2-phosphoglyceric acid, PEP, pyruvic acid, NADPH and ATP (Figure 2B). These findings are consistent with the results reported by Niedziela et al. [43] who observed that key enzymes of glycolysis were enhanced in Al-treated plants, leading to accelerated production of pyruvate and acetyl CoA.

Studies have shown that Al stress leads to energy deprivation in plant growth and development. As a result, tolerant plants accumulate high levels of glucose to produce adequate ATP through glycolysis [12,13,44]. For instance, in Al-tolerant rice, Wang et al. [14] showed that glycolytic pathway proteins are enhanced, which could maintain basic respiration needs and/or generate ATP for Al stress tolerance. A similar increase in glycolytic flux was noted in Al-tolerant citrus leaves [12]. The magnitude of glucose production in plants through the CBC is crucial in confirming Al stress tolerance [36]. Accumulation of glucose under Al stress also functions as an osmoprotectant, which decreases osmotic potential, regulates turgor dynamics and promotes the maintenance of membrane integrity and overall cellular homeostasis [7,45]. This osmoregulation enables plants to maintain sufficient carbohydrate reserves for sustaining essential metabolism under Al stress [36]. These findings indicate that the 1% PA treatment can trigger a metabolic switch from growth to survival mode via glucose metabolism for energy generation. Furthermore, biostimulants have been reported to enhance plant tolerance to various stresses [46]. Similar to many other biostimulants, PA has been reported to enhance the production of proteins and metabolites associated with carbohydrate metabolism and energy production [3,23]. Consistent with the study of Wang et al. [23], PA treatment increased the abundance of glycolytic metabolic pathway proteins, increasing plant tolerance to drought. Additionally, the bioactive compounds in PA may act as signaling molecules to regulate key metabolic pathways, including CCM, and/or stimulate differential gene expression to promote plant growth and resilience to Al stress [3,21,23]. Hence, increasing glycolytic metabolites with 1% PA treatment can be considered as an effective strategy to meet the energy demand for Al tolerance and adaptation.

#### 3.2.3. Pentose Phosphate Pathway

The PPP is a critical metabolic pathway for glucose degradation and is important for providing reducing power and intermediate metabolites for other pathways [47]. The present study showed that the total PPP metabolites were affected by both PA and Al treatment (Table 1; Figure 3A). The total PPP metabolites were reduced by approximately 0.23-fold and 0.3-fold with 0.25% PA and 1% PA, respectively, compared to the control. The decrease in total PPP metabolites in 1% PA-treated plants can be attributed to low levels of ribulose-5P and sedoheptulose-7P, although 6-phosphogluconate levels were higher compared to the control (Figure 3A). On the other hand, Al stress markedly reduced the total PPP metabolites, and PA treatment could not alleviate these effects compared to the control (Table 1). For instance, under 1 mM Al, the total PPP metabolites were substantially reduced by approximately 0.35-fold and 0.69-fold with 0.25% PA and 1% PA treatments, respectively, compared to 1 mM Al treatment alone. The reduction in total PPP metabolites in 0.25% PA-treated plants can be attributed to low glucose-6P, ribose-5P, glyceraldehyde-3P, fructose-6P, erythrose-4P and sedoheptulose-7P contents. Similarly, the reduction in total PPP metabolites following the application of 1% PA can be attributed to low glucose-6P, ribulose-5P, ribose 5-P, glyceraldehyde-3P, fructose-6P, erythrose-4P and sedoheptulose-7P contents, although 6-phosphogluconate levels were higher compared to the 1 mM Al treatment alone (Figure 3A).

The 4 mM Al significantly reduced the total PPP metabolites, which were not different from those of 0.25% and 1% PA-treated plants under the same conditions of Al stress (Table 1). Furthermore, the substantial reduction in total PPP metabolites in both control and PA-treated plants under 4 mM Al can be attributed to low levels of all PPP metabolites except 6-phosphogluconate and sedoheptulose-7P contents (Figure 3A). Nicotinamide adenine dinucleotide phosphate (NADPH) is the primary reducing power in the PPP pathway and is used to maintain cellular redox homeostasis via antioxidant production [47]. However, the levels of NADPH were not affected by PA treatment but were reduced under 4 mM Al stress (Figure 3A). It has been established that Al stress leads to ROS accumulation, which causes cellular damage and programmed cell death [48]. The results of this study indicate that a reduction in NADPH levels under Al stress could increase ROS generation and compromise the antioxidant defense system in plants.

#### 3.2.4. Tricarboxylic Acid Cycle

The TCA cycle is a central biochemical pathways for respiratory substrate oxidation and the production of ATP for all cellular functions [49]. In response to Al stress, the TCA cycle serves as a reservoir for organic acids (OA) production, which has been strongly linked to Al tolerance in most plants [2,4]. Our study revealed that the total TCA cycle intermediate metabolites were affected by PA and Al treatments (Table 1; Figure 3B). In the absence of Al stress, the total TCA cycle metabolites increased by approximately 0.24-fold and 0.46-fold with PA treatment compared to the control. The increased total TCA cycle metabolites with 0.25% PA could be due to increased levels of oxaloacetic acid, citric acid, succinic acid, fumaric acid and malic acid (Figure 3B). Likewise, the increased total TCA cycle metabolites with 1% PA treatment can be ascribed to high contents of all TCA cycle metabolites except for acetyl-CoA and aconitic acid, which were not altered compared to the control (Figure 3B). Similarly, in response to Al stress, PA-treated plants accumulated high levels of TCA cycle metabolites compared to plants treated with Al alone. Exposing the plants to 4 mM Al without PA increased the total TCA cycle metabolites by approximately 0.11-fold due to high levels of citric acid, aconitic acid, α-ketoglutaric acid and malic acid (Figure 3B). It has been established that primary metabolic constituents associated with Al tolerance are strongly linked to mitochondrial metabolism and OA production [2,4,8]. Organic acid production in response to Al stress is the most-characterized mechanism for Al tolerance, with citrate and malate being the common TCA cycle metabolites identified in several plants [4,8,13]. For instance, tomatoes produce and secrets malate, which chelate Al ions into non-toxic forms [50]. The increased OA production observed under Al stress in this study is consistent with previous studies in which mitochondrial citrate and malate levels were enhanced and shown to improve Al tolerance in tomato [50], maize (*Zea mays*) [8,13,15], buckwheat (*Fagopyrum esculentum*) [51], cabbage (*Brassica oleracea*) [52] and wheat (*Triticum aestivum*) [53].

Furthermore, both 0.25% PA and 1% PA treatments resulted in the accumulation of approximately 0.31-fold and 0.97-fold of the total TCA cycle metabolites, respectively, under 1 mM Al compared to plants treated with 1 mM Al alone. Under 4 mM Al, the total TCA cycle metabolites increased by approximately 0.03-fold and 0.41-fold in 0.25% PA and 1% PA-treated plants, respectively, compared to plants exposed to 4 mM Al alone (Table 1). The increase in total TCA cycle metabolites with 1% PA treatment under Al stress could be due to high levels of all TCA metabolites except for aconitic acid, which was reduced (Figure 3B). Similarly, Wang et al. [23] reported that PA treatment enhanced the TCA cycle in wheat seedlings under drought stress due to increased activities of malate dehydrogenase. Additionally, Sweetlove et al. [49] suggested that the activities of the different TCA cycle enzymes are independent of each other, and the TCA metabolites vary in their flux levels. Although the activities of these enzymes were not examined in this study, the increased TCA cycle metabolites following PA treatment could be associated with an increase in enzyme activities. This indicates that PA can promote the TCA cycle to produce sufficient ATP to accommodate the energy demand of plants under Al stress [23]. Evidence from numerous studies revealed that some plants accumulate high levels of TCA cycle metabolites internally to detoxify Al in roots and leaf cells and compartmentalize Al–OA complexes into the vacuole [2,51]. This internal Al complexation could prevent Al from interacting with macromolecules, including nucleic acids and proteins. Although this Al tolerance mechanism is mostly associated with hyperaccumulators [4,51], the increased TCA cycle metabolites in the leaves of both PA and control-treated plants under Al stress could suggest the stimulation of internal detoxification of Al ions, thereby promoting Al tolerance. Moreover, the TCA cycle metabolites are not only known to chelate Al ions. For instance, citrate plays an important role in antioxidant production and is involved in respiratory assimilation to produce energy for stress defense [54]. Additionally, α-ketoglutaric acid is critical in respiration and nitrogen metabolism for amino acid biosynthesis, which regulate osmotic potential and mediate stress tolerance in plants [55]. Hence, the increased TCA cycle metabolites in PA and control plants under Al stress can also be linked to amino acid and nucleic acid biosynthesis for Al tolerance.

#### 3.2.5. Electron Transport Chain (ETC)

The electron transport chain is one of the most critical pathways for both cellular respiration and photosynthesis [49]. It consists of an array of electron transporters that oxidize reducing equivalents for energy generation *via* ATP biosynthesis [11]. The results of this study revealed that PA and Al treatment altered the total ETC metabolites in tomato leaves (Table 1; Figure 4). The total ETC metabolites were increased by approximately 0.16-fold and 0.45-fold in 0.25% PA and 1% PA-treated plants, respectively, compared to the control in the absence of Al stress (Table 1). This increase in ETC metabolites following PA treatments could be due to high levels of FMN, FAD, NAD and NADH contents (Figure 4). Nonetheless, the levels of ADP and ATP remained unchanged with PA treatments (Figure 4). When exposed to Al stress, the tomato plants slightly reduced their total ETC metabolites by approximately 0.13-fold under 4 mM Al, while those under 1 mM Al exhibited no considerable change. The reduction in total ETC metabolites with 4 mM Al can be attributed to low ADP and ATP contents compared to the control (Figure 4). Accumulating evidence has revealed that Al stress impairs the activities of ETC complexes and could result in decreased metabolite production [38,56,57,58].

Moreover, both 0.25% and 1% PA treatments under 1 mM Al increased the total ETC metabolites by approximately 0.12-fold and 0.31-fold, respectively, compared to plants exposed to 1 mM Al. The increased total ETC metabolites in the 0.25% PA-treated plants can be associated with increased FMN, FAD, NAD and ATP contents, while that of 1% PA-treated plants is due to high FMN, FAD, NAD, NADH and ATP contents (Figure 4). However, the total ETC metabolites were slightly increased in plants treated with 0.25% PA under 4 mM Al, while 1% PA decreased the total ETC metabolites compared to 4 mM Al alone (Table 1).

Additionally, the increase in ETC metabolites with 0.25% PA can be attributed to a slight increase in FMN, FAD, NAD and NADH contents, while the reduction in ETC metabolites with 1% PA treatments was due to low NADH contents (Figure 4). ATP synthase is critical for ATP production in ETC [11]. According to Su et al. [58], Al stress targets and damage subunits of ATP synthase, thereby reducing the amount of ATP generated. With a limitation in ATP production and electron transport, electrons leakage is enhanced, which could result in uncontrollable ROS generation and oxidative stress-induced cell death [11,57]. Interestingly, Wang et al. [23] reported that the relative abundance of proteins related to ATP synthesis was increased in PA-treated wheat seedlings under drought stress. although ATP-related proteins were not examined in this study, the increase in ATP production in PA-treated plants alone and with 1 mM Al stress partially aligns with the findings of Wang et al. [23]. In plant stress adaptation, the energy cost is high, and the enhancement of ETC processes with PA treatments suggests that restoring energy supply via ATP production is critical to protect several basal metabolic processes under Al stress.

### 3.3. Associations between Central Carbon Metabolites

To further examine the association among the intermediate metabolites of the CCM routes in the PA-treated tomato seedlings under Al stress, a 2D PCA and Pearson correlation were used (Table 2; Figure 5 and Figure 6). The Pearson correlation analysis showed that the total CBC metabolites had a significantly (*p* < 0.001) strong positive association with the total PPP metabolites and a weak negative association with glycolysis and TCA cycle metabolites (Table 2). Additionally, the total glycolytic metabolites had a significantly (*p* < 0.05) moderate positive correlation with total TCA cycle metabolites and a weak negative association with total PPP and ETC metabolites (Table 2). These findings are consistent with the results reported by Niedziela et al. [43], who demonstrated that key enzymes of glycolysis were enhanced in Al-treated plants, leading to accelerated pyruvate and acetyl CoA production for OA synthesis. These results suggest that energy production plays a crucial role in supporting basic cellular function under Al stress, and that plants channel their photosynthetic carbon for stress adaption [14]. On the other hand, total PPP metabolites exhibited non-significant (*p* > 0.05) weak negative and positive associations with the total TCA cycle and ETC metabolites, respectively (Table 2). Similarly, the total TCA cycle metabolites had a weak positive correlation with the total ETC metabolites. The increase in glycolytic flux under Al stress proved that carbon skeletons produced via glycolysis are used primarily for OA production to detoxify Al ions, thereby promoting Al tolerance [13,50,51].

Although some of the total metabolites showed a non-significant (*p* > 0.05) weak correlation with other metabolites, the CBC, glycolysis and PPP shared common metabolites which showed significant (*p* < 0.05) associations (Figure 5; Appendix A). For example, RuBP had a significantly (*p* < 0.01) strong positive association with fructose 1,6BP and ATP, and a moderate positive association with glucose-6P and xylulose-5P (Figure 5; Appendix A). Similarly, sedoheptulose-7BP showed a significantly (*p* < 0.01) moderate positive correlation with fructose-6P, erythrose-4P and glucose-6P, and a non-significant (*p* > 0.05) moderate positive association with ATP and xylulose-5P. Additionally, both ribose-5P and ribulose-5P had a significantly (*p* < 0.01) strong positive association with erythrose-4P and glucose-6P, and a moderate association with fructose-6P. However, both sedoheptulose-7P and ribulose-5P exhibited a significantly (*p* < 0.01) strong negative association with pyruvic acid, while ribulose-5P had a moderate association with xylulose-5P (Figure 5; Appendix A). Moreover, the correlation analysis among the individual metabolites confirmed that the glycolysis intermediates had a significant (*p* < 0.05) association with the TCA cycle intermediate metabolites (Figure 5; Appendix A). Fructose-1,6BP had a strong positive association with succinic acid and fumaric acid and a negative association with aconitic acid. Glucose had a strong association with isocitric acid, α-ketoglutaric acid and succinic acid, and a moderate association with oxaloacetic acid and citric acid (Figure 5; Appendix A). Similarly, pyruvic acid had a strong positive association with oxaloacetic acid, citric acid, isocitric acid and α-ketoglutaric acid, and a moderate positive association with succinic acid and acetyl-CoA (Figure 5; Appendix A).

A 2D PCA biplot revealed a projection of response variables in the factor spaces (F1 and F2) and explained approximately 82% of the total disparity in the dataset (Figure 6). The control treatment exhibited a strong influence on both the CBC and PPP, and a moderate influence on ETC. Similarly, both 0.25% PA and 1% PA showed a strong influence on ETC and a moderate influence on the CBC and PPP (Figure 6). However, under Al stress, 0.25% PA and 1 mM Al moderately influenced the CBC and PPP, while 1% PA showed a strong influence on TCA and glycolysis. Additionally, 1% PA under 4 mM Al showed a moderate influence on glycolysis and TCA, while both 0.25% PA under 4 mM Al and 4 mM alone had no influence on the CCM metabolites (Figure 6). These findings are consistent with the findings of Wang et al. [23], who reported that PA promotes the glycolysis pathway, which fuels the TCA cycle to produce sufficient energy for stress tolerance. In summary, the coordinated association between the determined metabolites confirmed that PA stimulates alterations in plant metabolism to increase energy production and organic acids biosynthesis for Al stress tolerance (Figure 7).

## 4. Conclusions

Acidic soils are widespread, and Al stress is a major limiting factor for plant growth and productivity in acidic soils. In the present study, we comprehensively investigated the metabolic response of key intermediate metabolites in central carbon metabolism routes in PA-treated tomato seedlings under Al stress. A total of 48 differentially expressed metabolites involved in the Calvin–Benson cycle, glycolysis, pentose phosphate pathway, tricarboxylic acid cycle and electron transport chain were identified in the leaves of both control and PA-treated tomato plants under Al stress. Aluminum stress considerably reduced the levels of these metabolites, while PA treatment triggered dynamic metabolic alterations and played an important role in Al stress adaptation. The coordinated association among these identified metabolites revealed that PA stimulates changes in plant metabolism to modulate energy production and the biosynthesis of organic acids for Al stress tolerance. However, further investigation is required to examine how PA treatment influences the genes and enzymes involved in these CCM routes under Al stress.

## Figures and Tables

**Figure 1 metabolites-13-00770-f001:**
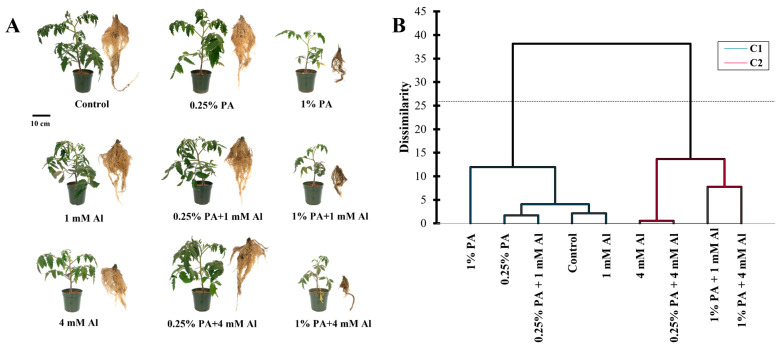
Response of Scotia tomato seedlings to pyroligneous acid (PA) treatment under aluminum stress (Al). (**A**) Morphological effect on tomato growth. (**B**) Cluster analysis of overall central carbon metabolite composition in the leaves.

**Figure 2 metabolites-13-00770-f002:**
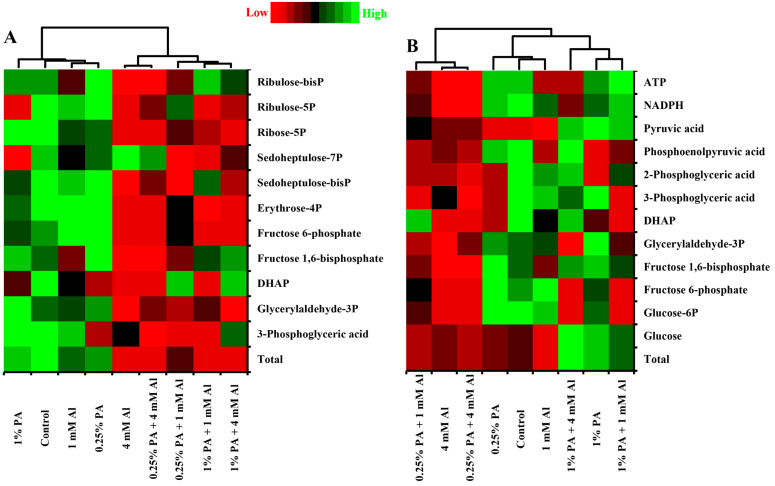
Heat map of metabolites involved in (**A**) the Calvin–Benson cycle and (**B**) the glycolysis pathway in leaves of Scotia tomato seedlings treated with pyroligneous acid (PA) under aluminum (Al) stress. Metabolite concentrations in each compartment are normalized across all data for an individual compound such that similar colour intensities between compounds can represent widely differing concentrations. The red colour represents a lower concentration, and the green colour represents a higher concentration of a particular metabolite.

**Figure 3 metabolites-13-00770-f003:**
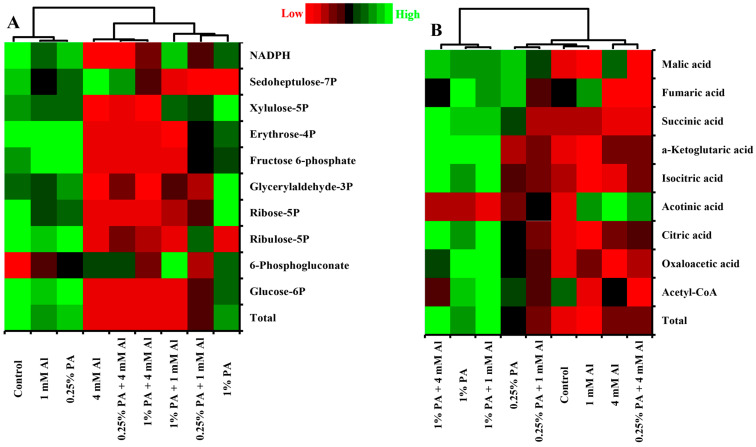
Heat map of metabolites profile involved in (**A**) pentose phosphate pathway and (**B**) tricarboxylic acid (TCA) cycle in leaves of Scotia tomato seedlings treated with pyroligneous acid (PA) under aluminum (Al) stress. Metabolite concentrations in every compartment are normalized across all data for an individual compound such that similar colour intensities between compounds can represent widely differing concentrations. The red colour represents a lower concentration, and the green colour represents a higher concentration of a particular metabolite.

**Figure 4 metabolites-13-00770-f004:**
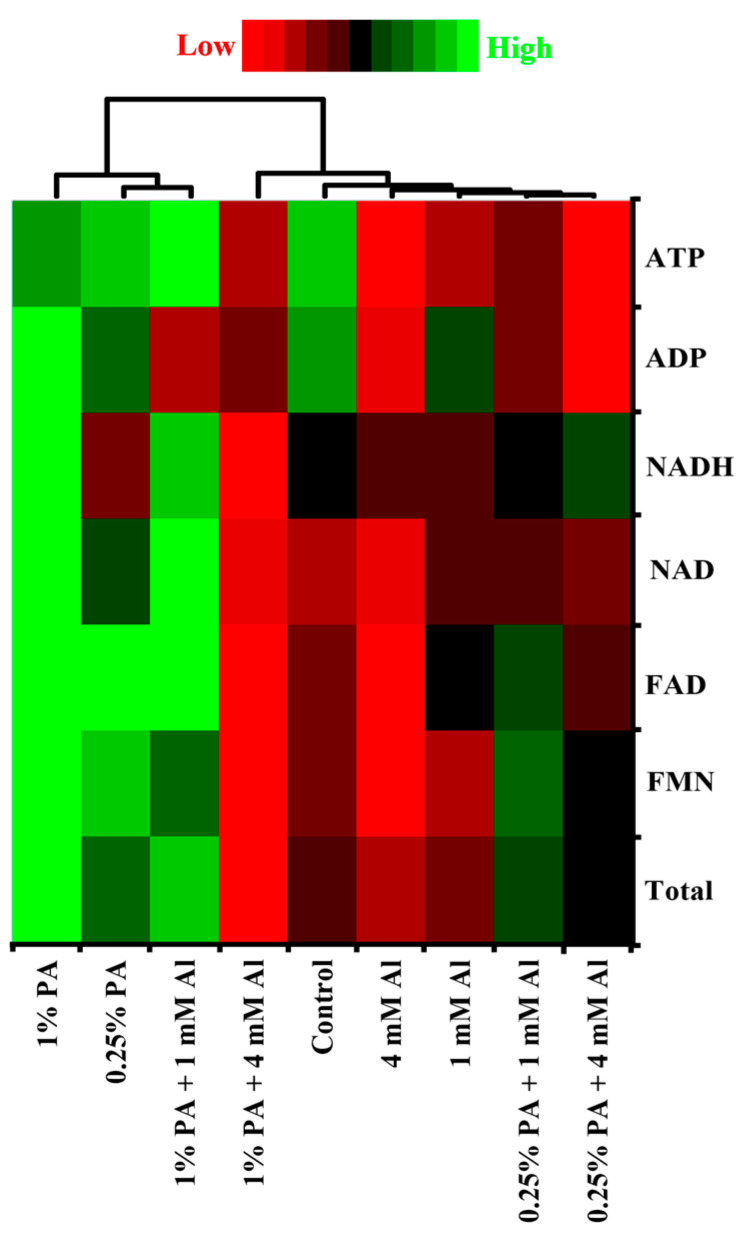
Heat map of metabolites profile involved in electron transport chain in the leaves of Scotia tomato seedlings treated with pyroligneous acid (PA) under aluminum (Al) stress. Metabolite concentrations in each compartment are normalized across all data for an individual compound, such that similar colour intensities between compounds can represent widely differing concentrations. The red colour represents a lower concentration, and the green colour represents a higher concentration of a particular metabolite.

**Figure 5 metabolites-13-00770-f005:**
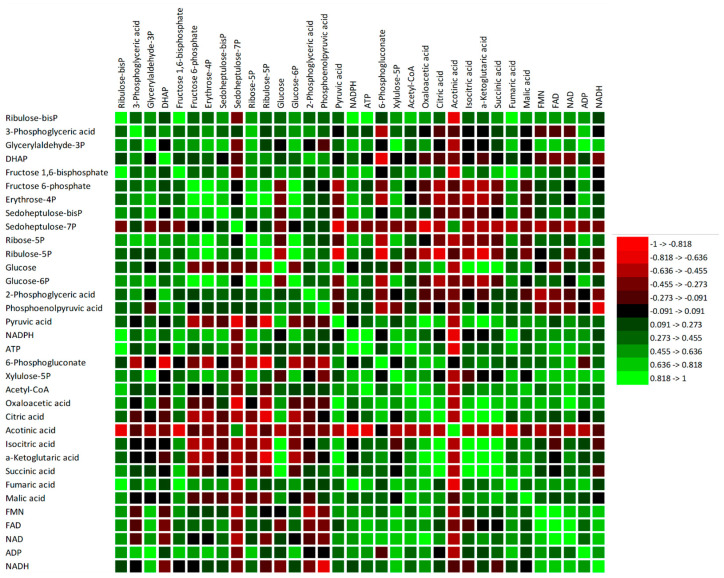
Pearson correlation matrix among individual metabolites of the central carbon metabolic pathway in Scotia tomato seedlings treated with pyroligneous acid (PA) under aluminum (Al) stress. The red colour represents a strong negative association, and the green colour represents a strong positive association. DHAP, dihydroxyacetone phosphate; NADP, nicotinamide adenine dinucleotide phosphate; FMN, flavin mononucleotides; FAD, flavin adenine dinucleotide; ADP, adenosine diphosphate; ATP, adenosine triphosphate.

**Figure 6 metabolites-13-00770-f006:**
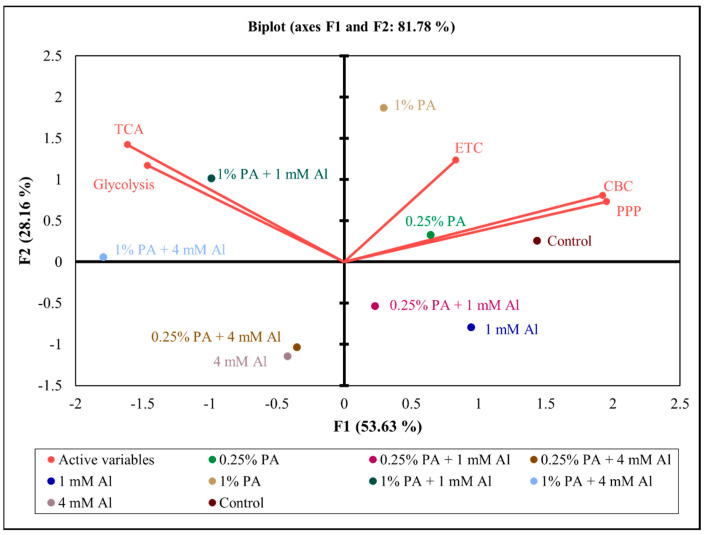
A two-dimensional principal component analysis (2-D PCA) biplot showing relationships amongst the explanatory variables (total metabolites involved in specific central carbon metabolic pathways) Calvin–Benson cycle (CBC), glycolysis, pentose phosphate pathway (PPP), tricarboxylic acid (TCA) cycle and electron transport chain (ETC) of Scotia tomato seedlings treated with pyroligneous acid (PA) under aluminum (Al) stress. The projection of the variables in the 2-D factor space (F1 and F2) explained a total of 81.78% of the variations in the dataset. Variables that are closely located are not different compared to variables located at a distance within a quadrant or between quadrants.

**Figure 7 metabolites-13-00770-f007:**
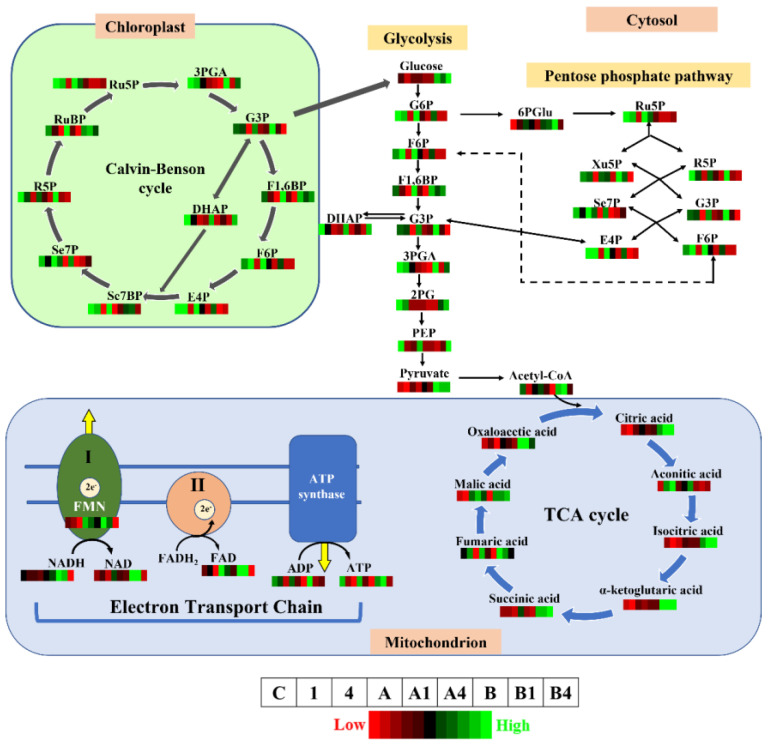
Overview of metabolic changes of key central carbon pathways in the leaves of Scotia tomato seedlings treated with pyroligneous acid (PA) under aluminum (Al) stress. Ru5P, ribulose-5-phosphate; 3PGA, 3-phosphoglyceric acid; G3P, glyceraldehyde-3-phosphate; F1,6BP, fructose-1,6-bisphosphate; F6P, fructose-6-phosphate; E4P, erythrose-4-phosphate; Se7BP, sedoheptulose-1,7-bisphosphate; Se7P, sedoheptulose-7-phosphate; R5P, ribose-5-phosphate; RuBP, ribulose-1,5-bisphosphate; G6P, glucose-6-phosphate; DHAP, dihydroxyacetone phosphate; 2PG, 2-phosphoglyceric acid; PEP, phosphoenolpyruvic acid; 6PGlu, 6-phosphogluconate; Xu5P, xylulose-5-phosphate; NADP, nicotinamide adenine dinucleotide phosphate; FMN, flavin mono-nucleotides; FAD, flavin adenine dinucleotide; ADP, adenosine diphosphate; ATP, adenosine triphosphate. The red colour of the metabolic pattern represents a lower concentration, and the green colour represents a higher concentration of a particular metabolite. The treatments that match the corresponding colour patterns were arranged from left to right as control (C), 1 mM Al (1), 4 mM Al (4), 0.25% PA (A), 0.25% PA + 1 mM Al (A1), 0.25% PA + 4 mM Al (A4), 1% PA (B), 1% PA + 1 mM Al (B1) and 1% PA + 4 mM Al (B4).

**Table 1 metabolites-13-00770-t001:** Total metabolites involved in central carbon metabolic pathways in the leaves of tomato (*Solanum lycopersicum*, ‘Scotia’) seedlings treated with pyroligneous acid (PA) under aluminum (Al) stress.

Treatment	CBC (nmol g^−1^ FW)	Glycolysis (µmol g^−1^ FW)	PPP (nmol g^−1^ FW)	TCA (µmol g^−1^ FW)	ETC (nmol g^−1^ FW)
Control	23.79	21.72	30.01	6.97	5.63
1 mM Al	15.54	15.18	20.70	6.08	5.38
4 mM Al	6.10	20.63	5.97	7.77	4.88
0.25% PA	16.15	20.72	22.90	8.66	6.51
0.25% PA + 1 mM Al	11.10	15.81	13.40	7.99	6.05
0.25% PA + 4 mM Al	5.30	15.60	5.23	8.03	5.75
1% PA	17.18	38.16	20.97	10.18	8.17
1% PA + 1 mM Al	6.50	32.36	6.33	11.97	7.07
1% PA + 4 mM Al	5.33	56.65	5.13	10.92	3.53
CV (%)	55.70	52.56	64.72	21.97	22.43

CBC, Calvin–Benson cycle; PPP, pentose phosphate pathway; TCA, tricarboxylic acid cycle; ETC, electron transport chain.

**Table 2 metabolites-13-00770-t002:** Pearson correlation coefficients (r) amongst the specific central carbon metabolic pathways in Scotia tomato seedlings treated with pyroligneous acid (PA) under aluminum (Al) stress and their corresponding significance levels at *p* ≤ 0.05.

Variables	CBC	Glycolysis	PPP	TCA
Glycolysis	r = −0.240 *p* = 0.533			
PPP	r = 0.993 *p* = 0.000	r = −0.274 *p* = 0.475		
TCA	r = −0.436 *p* = 0.240	r = 0.762 *p* = 0.017	r = −0.461 *p* = 0.211	
ETC	r = 0.319 *p* = 0.403	r = −0.303 *p* = 0.429	r = 0.302 *p* = 0.430	r = 0.147 *p* = 0.706

CBC, Calvin–Benson cycle; PPP, pentose phosphate pathway; TCA, tricarboxylic acid cycle.

## Data Availability

The data presented in this study are available from the corresponding author upon request. Data is not publicly available due to privacy or ethical restrictions.

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
