# Peer review of "Coordinated Regulation of Central Carbon Metabolism in Pyroligneous Acid-Treated Tomato Plants under Aluminum Stress"

_metabolites, 2023, doi:10.3390/metabo13060770_

Round 1

Reviewer 1 Report

The authors of the manuscript “Coordinated regulation of central carbon metabolism in pyroligneous acid-treated tomato plants under aluminum stress” evaluated the changes induced in tomato metabolites by pyroligneous acid-treated seedlings under aluminum stress using LC_MRM_MS. The results are encouraging. The authors also proposed a model for changes in key central carbon pathways in tomato plants treated with pyroligneous acid under aluminum stress. However, the authors failed to provide metabolite-dependent MS parameters used in MRM, linearity of calibration, detection and quantification limit of studied metabolites. Also, the authors should provide the content of each metabolite identified from the leaf tissues of tomato plants.

L141: Leave tissues (15 leaves per treatment). Please indicate the position of the leaves. How many plants are in each treatment?

L217: Did the authors observe the phytotoxicity (morphology) effects of PA and Al on tomato plants? If so, please provide data.

L164, 177 & 188: Please provide specific metabolite-dependent MS parameters used in multiple reaction monitoring.

L250: Please indicate the limit of detection and quantification of studied metabolites.

Please provide the chromatogram of each metabolite.

Please indicate the metabolite content from tomato leaf tissues analyzed by LC-MRM-MS.

Minor editing of the English language required

Author Response

L141: Leave tissues (15 leaves per treatment). Please indicate the position of the leaves. How many plants are in each treatment?

Revised. Also, we had five replications per treatment (Line 137)

L217: Did the authors observe the phytotoxicity (morphology) effects of PA and Al on tomato plants? If so, please provide data.

Revised add a photograph of the morphological effect. However, the morpho-physiological and biochemical response of PA-treated tomato under Al stress has been submitted to a different journal for publication.

L164, 177 & 188: Please provide specific metabolite-dependent MS parameters used in multiple reaction monitoring.

Revised

L250: Please indicate the limit of detection and quantification of studied metabolites. Please provide the chromatogram of each metabolite.

Revised and the chromatogram of each metabolite has been added to the supplementary figures.

Please indicate the metabolite content from tomato leaf tissues analyzed by LC-MRM-MS.

Please each identified metabolite contents were analysed and presented in the Heatmaps.

Reviewer 2 Report

The manuscript is very interesting, it concerns metabolic changes  occurring in biostimulated plants (tomato plants treated with pyroligneous acid) exposed to aluminium stress. Therefore, the subject is important and up-to-date, since the possibility of the increased tolerance to stresses induced by biostimulants is a subject of much study.

The Authors performed well designed experiments and analyses, resulting in the identification of slmost 50 metabolites involved in the Calvin-Benson cycle, glycolysis, pentose phosphate pathway, tricarboxylic acid cycle and electron transport chain. Aluminum stress reduced the levels of these metabolites while pyroligneous acid treatment triggered various metabolic changes that might play an important role in adaptation to alumiium stress. The manuscript is illustrated with heat-maps, cluster and principal component analyses, and summarized with final overview of metabolic changes of key central carbon pathways.

In my opinion, the manuscript is valuable and can be published after minor revision, involving small changes to clarify the text.

Line 88. “PA is a complex mixture of over 200 water-soluble bioactive compounds including organic acids, phenolics, sugar derivatives and alcohol” - please precise what is “alcohol” in this context, ethanol, methanol or other compounds (in chemistry alcohol means an organic compound in which a hydroxyl group is bound to a carbon atom).

Line 90. “PA has been demonstrated to promote seed germination, vegetative and yield of several crops” – something seems to be missing in this sentence – vegetative what?

Line 97. „increased enzymes” – please precise, if it means an increase in enzyme activity or accumulation (increased biosynthesis)? These are two different factors.

Line 114. Please insert the Latin name of white pine (I guess that „white pine” is Pinus strobus L.), otherwise “white pine biomass” can be mistakenly understood.

LIne 119. 5 min (not 5 mins)

Author Response

Line 88. “PA is a complex mixture of over 200 water-soluble bioactive compounds including organic acids, phenolics, sugar derivatives and alcohol” - please precise what is “alcohol” in this context, ethanol, methanol or other compounds (in chemistry alcohol means an organic compound in which a hydroxyl group is bound to a carbon atom).

Revised to alcohol derivatives.

Line 90. “PA has been demonstrated to promote seed germination, vegetative and yield of several crops” – something seems to be missing in this sentence – vegetative what?

Revised

Line 97. „increased enzymes” – please precise, if it means an increase in enzyme activity or accumulation (increased biosynthesis)? These are two different factors.

Revised

Line 114. Please insert the Latin name of white pine (I guess that „white pine” is Pinus strobus L.), otherwise “white pine biomass” can be mistakenly understood.

Revised

Round 2

Reviewer 1 Report

The authors failed to provide metabolite-dependent MS parameters used in MRM, linearity of calibration, detection, and quantification limit of studied metabolites.

Also, the authors should provide the content of each metabolite identified from the leaf tissues of tomato plants.

Com: Please indicate the metabolite content from tomato leaf tissues analyzed by LC-MRM-MS.

Res: Please each identified metabolite contents were analysed and presented in the Heatmaps

Please provide the contents (µmol g−1 FW or nmol g−1 FW). It is hard to find the exact content of each metabolite presented in Heatmaps. 

Minor text editing is required.

Author Response

The authors failed to provide metabolite-dependent MS parameters used in MRM, linearity of calibration, detection, and quantification limit of studied metabolites.

Thank you for the comments. Actually, the metabolite-dependent MS parameters used in MRM including linearity of calibration, detection, and quantification limit of studied metabolites have been cited in the materials and methods which do not require us to repeat them. Please refer to the methods described by Han et al. (2013a) (doi: 10.1002/elps.201200601) and Han et al. (2013b) (doi: 10.1021/ac400769g) for further information. Additionally, the leaf samples were sent out to one of the renowned institutions in Canada, The UVic-Genome BC Proteomics Centre (PC), for the metabolite analysis as indicated in our materials and methods section. The PC is the central hub of the Pan-Canadian Proteomics Centre, Genome Canada’s Genomics Technology Platforms for proteomics technology development and service. In addition, the PC is a founding member of The Metabolomics Innovation Centre, Canada’s leading metabolomics facility, focused on quantitative metabolomics.  The Centre has been providing mass spectrometry analytical services to academic, industrial, and government laboratories for over 30 years.

Also, the authors should provide the content of each metabolite identified from the leaf tissues of tomato plants.

Please indicate the metabolite content from tomato leaf tissues analyzed by LC-MRM-

Thank you for the comments. We have added the content of each metabolite to the supplementary table (Table S1).

Round 3

Reviewer 1 Report

Actually, the metabolite-dependent MS parameters used in MRM including linearity of calibration, detection, and quantification limit of studied metabolites have been cited in the materials and methods which do not require us to repeat them. 

The metabolite-dependent MS parameters like the linearity of calibration, detection, and quantification limit of studied metabolites always varied. Please recheck with UVic-Genome BC Proteomics Centre.

 Minor editing of English language required